# The Characterization of the Vertical Distribution of Surface Soil Moisture Using ISMN Multilayer In Situ Data and Their Comparison with SMOS and SMAP Soil Moisture Products

**Na Yang** [1,*] **, Feng Xiang** [2] **and Hengjie Zhang** [1]

1 School of Surveying and Land Information Engineering, Henan Polytechnic University, Jiaozuo 454000, China; zhanghengjie123@foxmail.com
2 College of Computer Science and Technology, Henan Polytechnic University, Jiaozuo 454000, China; xiangfeng821008@foxmail.com
* Correspondence: yangna@hpu.edu.cn; Tel.: +86-0391-3987661

**Abstract:** In this paper, we investigated the vertical distribution characteristics of surface soil moisture based on ISMN (International Soil Moisture Network) multilayer in situ data (5, 10, and 20 cm; 2, 4, and 8 in) and performed comparisons between the in situ data and four microwave satellite remote sensing products (SMOS L2, SMOS-IC, SMAP L2, and SMAP L4). The results showed that the mean soil moisture difference between layers can be $-0.042 \sim -0.024$ (for the centimeter group)/$-0.067 \sim -0.044$ (for the inch group) m³/m³ in negative terms and 0.020~0.028 (for the centimeter group)/0.036~0.040 (for the inch group) m³/m³ in positive terms. The surface soil moisture was found to have very significant stratification characteristics, and the interlayer difference was close to or beyond the SMOS and SMAP 0.04 m³/m³ nominal retrieval accuracy. Comparisons revealed that the satellite retrievals had a higher correlation with the field measurements of 5 cm/2 in, and SMAP L4 had the smallest difference with the in situ data. The mean difference caused by using 10 cm/4 in and 20 cm/8 in situ data instead of the 5 cm/2 in data could be about $-0.019 \sim -0.018$/$-0.18 \sim -0.015$ m³/m³ and $-0.026 \sim -0.023$/$-0.043 \sim -0.039$ m³/m³, respectively, meaning that there would be a potential depth mismatch in the data validation.

**Keywords:** soil moisture; calibration and validation; Soil Moisture and Ocean Salinity (SMOS); Soil Moisture Active Passive (SMAP)

## 1. Introduction

The SMOS (Soil Moisture and Ocean Salinity, ESA, November 2009) and SMAP (Soil Moisture Active Passive, NASA, January 2015) missions are dedicated to the acquisition of global soil moisture information. They both use the L band (1.4/1.41 GHz) in the mode of passive microwave remote sensing, as there would be a greater depth of penetration due to the longer wavelength [1,2]. The soil moisture products (retrievals and estimates) nominally released by SMOS and SMAP are the average soil moisture at the top of the surface, and they are conventionally compared with 5 cm in situ data [3,4]. However, the response depth of the L band is likely to vary from a very thin surface to a certain deep layer due to the variety and instability of the observing conditions in practice, which are difficult or impossible to measure accurately at present [5–7]. The depth mismatch would potentially be present in the comparisons of SMOS and SMAP soil moisture products and also in their comparisons with soil moisture field measurements, which is commonly thought to introduce uncertainties in the validation of multisource data [8–17].

From a data flow perspective, the soil moisture products from SMOS and SMAP can be considered the comprehensive results of three main processes, namely, brightness temperature (TB) observation, brightness temperature simulation, and soil moisture retrieval [18–23]. The numerical difference presented in validation and comparison can, in

this context, consist of two parts. The first one would be collectively called the "retrieval error" and may be caused by upstream phases, including the instruments' performances, observing conditions, reconstruction methods, radiative transfer models and parameter settings, auxiliary information inputs, and iterative computational strategies [24–29]. The other one is generally referred to as the "verification uncertainty" and is mainly caused by the difference in scale, depth, and time between the multisource data [30–33]. To some extent, in order to accurately find out the source of the "retrieval error", further understand its propagation mechanism, and make corresponding improvements, one should first exclude the "verification uncertainty" due to the spatial and temporal mismatch of the multisource data; in other words, they must adopt a way of tracing back from the downstream stage to the upstream stage, which is exactly the opposite of the flow of data production.

Based on high-frequency in situ measurements, the soil moisture at 5 cm undergoes natural fading of a very small magnitude during the time intervals between SMOS and SMAP, with an average variation (0.003 $m^3/m^3$ minimum; 0.007 $m^3/m^3$ maximum), that is insufficient to be identified using satellites (nominal accuracy 0.04 $m^3/m^3$), and the temporal mismatch may not cause external uncertainty and is negligible in data validation [34]. Similarly, by using multilayer in situ data as a reference, the effect of depth mismatch on the validation of SMOS and SMAP soil moisture products can be assessed to some extent. This paper attempts to make comparisons between L band microwave remote sensing soil moisture products and in situ soil moisture measurements, and the main objectives are as follows:

- To investigate the vertical distribution characteristics of surface soil moisture, the numerical characteristics of each layer, and the similarities and differences between the layers;
- To quantify the numerical difference between satellite soil moisture retrievals and multilayer in situ measurements;
- To demonstrate the effect of the depth mismatch, the rationality of using in situ data at one depth as a reference, and the feasibility of using another depth as a substitute.

## 2. Materials and Methods

### 2.1. Data

Five datasets were selected, and the time span was set to 1 January 2015~31 December 2020. The ISMN (International Soil Moisture Network) provides multilayer in situ soil moisture measurements, which were used to study the stratification characteristics and as a reference for the comparison with satellite products. SMOS L2 and SMAP L2 (passive) products are soil moisture retrievals; SMOS-IC and SMAP L4 products can be considered independent retrievals and estimates, respectively.

#### 2.1.1. ISMN In Situ Soil Moisture Data

The ISMN is an international collaboration to establish and maintain a global in situ soil moisture database. It brings together in situ soil moisture measurements collected and freely shared by a variety of organizations, harmonizes them in terms of units and sampling rates, applies advanced quality control, and stores them in a database [35,36]. In addition to single/multilayer soil moisture, static information (land cover, clay fraction, sand fraction, etc.) and other dynamic variables (soil temperature, air temperature, precipitation, etc.) are also included in the ISMN datasets. In general, soil moisture is quantified in terms of volumetric water content ($m^3/m^3$) and an hourly sampling rate.

#### 2.1.2. SMOS L2 Soil Moisture Product

The SMOS L2 Soil Moisture User Data Product (MIR_SMUDP2) consists of swath-based retrieved information over land surfaces. The base product includes fields for soil moisture, vegetation water content, calculated brightness temperatures at 42.5 °C, and dielectric constant from pole to pole. The product is organized in the form of a Discrete Global Grid (DGG) in the ISEA 4H9 (Icosahedral Snyder Equal Area) grid projection, and

the average distance between nodes is close to 15 km. The soil moisture retrievals (field: Soil_Moisture) are volumetric water content in $m^3/m^3$, and the accuracy requirement is set to 0.04 $m^3/m^3$ (i.e., 4% volumetric soil moisture) or better [37,38].

### 2.1.3. SMOS-IC Soil Moisture Product

The SMOS INRA-CESBIO (SMOS-IC) product provides global daily soil moisture and L band vegetation optical depth (L-VOD) from the ascending and descending orbits at a spatial resolution of 25 km (EASE grid 2.0). The SMOS-IC corresponds to the SMOS "original algorithm"; it is to be as independent as possible from auxiliary data, thus avoiding circular evaluation/validation [39]. The soil moisture retrievals (field: Soil_Moisture) are released in $m^3/m^3$ and with a dry bias of ~$-0.045$ $m^3/m^3$ against ISMN in situ sites [40]. The SMOS-IC V2 soil moisture product is the latest release (January 2020), and comparisons with in situ measurements and other "official" satellite products may help to better understand its characteristics.

### 2.1.4. SMAP L2 Soil Moisture Product

The SMAP L2 Radiometer Half-Orbit 36 km EASE-Grid Soil Moisture (L2_SM_P) product contains gridded data of passive soil moisture retrievals (in the top 5 cm of the soil column), ancillary data, and quality assessment flags on the 36 km global cylindrical Equal-Area Scalable Earth (EASE) Grid 2.0 projection and is presented in half-orbit granules. The soil moisture retrievals (field: Soil_Moisture) are volumetric water content in $m^3/m^3$, with an accuracy requirement of ~$\pm$0.04 $m^3/m^3$ [41,42].

Attention needs to be paid to the SMOS L2 and SMAP L2 soil moisture products. They are the direct retrieval outputs with Level 1 (L1) instrument brightness temperature observations as the input, and also the inputs used to generate Level 3 (L3) global daily soil moisture composites. The L2 products inherit the location and time codes of the L1 products but do not undergo the spatiotemporal resampling of the L3 products, thus avoiding the uncertainties introduced by data processing and ensuring reverse traceability from data validation to error location. For this reason, SMOS and SMAP L2 soil moisture products were used in this paper.

### 2.1.5. SMAP L4 Soil Moisture Product

The SMAP L4 Global 3-hourly 9 km EASE-Grid Surface and Root Zone Soil Moisture Geophysical Data (SPL4SMGP) contains global estimates of surface soil moisture (0–5 cm vertical average), root zone soil moisture (0–100 cm vertical average), and additional research products (soil temperature, evapotranspiration, etc.), based on the assimilation of SMAP L band brightness temperatures. This product appears on the EASE-Grid 2.0 projection at 9 km grid resolution, the soil moisture estimates (field: SM_Surface) are 3-hourly time-averaged volumetric water content in $m^3/m^3$, and the accuracy requirement is 0.04 $m^3/m^3$ [43,44]. It should be noted that SMOS also provides the L4 soil moisture product, but the coverage is limited to European and Mediterranean countries and therefore could not be used in this research.

The SMAP L4 soil moisture product has greater temporal continuity and spatial integrity than the L2 soil moisture product and is more application-oriented. The L4 product is formally at a higher level in the data system because it has more added value, but it is equivalent to the L2 product in terms of the data collection process because they both use the L1 product as input. The L2 and L4 products represent the two main ways of obtaining soil moisture information from satellite remote sensing; they reflect different implementation concepts, calculation methods, and spatiotemporal visualization systems, but both need to be verified and evaluated. It is therefore worth including the SMAP L4 soil moisture product in this study.

*2.2. Methods*

There are four parts to this section: the quality control of ISMN multilayer in situ data; the spatial and temporal matching of SMOS, SMAP products, and ISMN data; methods for the analysis of the stratification properties of soil moisture; and methods for the verification of the depth mismatch.

2.2.1. Quality Control of the In Situ Data

The ISMN in situ data of 1871 sites from 42 networks met the download conditions (global, 1 January 2015~31 December 2020). Although discussions on the accuracy and reliability of the data are beyond the scope of this article, quality control is still required. Following the three-level hierarchy of ISMN data storage, from network to site to variable file, the quality requirements were set as follows: First, networks with more than 10 sites should be retained. Second, sites should be selected that can provide 5, 10, and 20 cm soil moisture as well as 5, 10, and 20 cm soil temperature, i.e., there were 6 variables (must but not limited to) and only one sensor per depth (no multiple observations). It should be noted that some sites set the observation depth at 2 in, 4 in, and 8 in, which after conversion are 5.08 cm, 10.16 cm, and 20.32 cm respectively; such sites are also reserved as long as they have the six variables. Third, for each record (once per hour) in the variable file, it is considered "valid" if the 6 variables are all marked with "G" (good, ISMN Quality Flag), the number of such records should exceed 50% per year and every year from 2015 to 2020. In the end, 83 sites from 3 networks passed the quality check. The 3 networks are USCRN (U.S. Climate Reference Network), SCAN (Soil Climate Analysis Network), and SNOTEL (Snow Telemetry), and all 83 sites are located within the continental U.S., as shown in Figure 1A. Information on land cover, sand fraction, and clay fraction was read from the static variables file, as shown in Figure 1B.

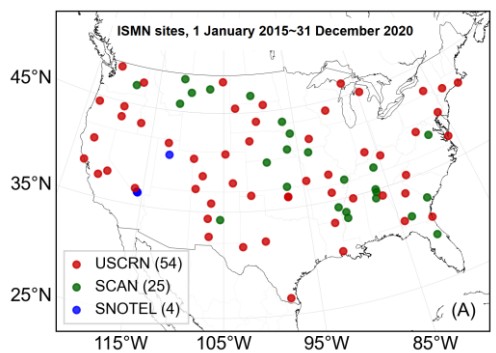 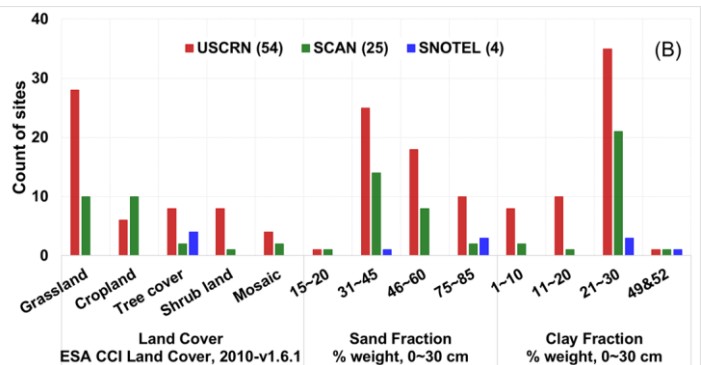

**Figure 1.** The 83 sites that passed the quality control: (**A**) spatial distribution of the sites; (**B**) information on land cover and soil properties of the sites.

2.2.2. Spatiotemporal Matching of In Situ Data and Satellite Products

Discussions on the retrieval and estimation accuracy of satellite products are beyond the scope of this article, and thus only the comparative differences between satellite soil moisture and multilayer in situ soil moisture were examined. As the data have different spatial and temporal characteristics, they had to be matched before performing any comparison.

The first step was spatial matching. The 83 sites from the 3 networks (USCRN, SCAN, and SNOTEL) provide hourly multilayer in situ soil moisture measurements; their locations are marked by longitude and latitude and are usually thought of as points in space. SMOS L2, SMOS-IC, SMAP L2, and SMAP L4 products are mapped in the ISEA 4H9 (~15 km), EASE-Grid 2.0–25 km, EASE-Grid 2.0–36 km, and EASE-Grid 2.0–9 km systems, respectively; the grids correspond to a specific area in space; and only the latitude and longitude of the grid center are given. The spatial matching of satellite products and in situ data can be performed according to the principle of closest distance. Taking the location of

each site as a reference, a five-element matching group (ISMN site, SMOS L2 grid center, SMOS-IC grid center, SMAP L2 grid center, and SMAP L4 grid center) can be formed to search separately for the satellite grid center that is closest to the site.

It should be noted that no horizontal rescaling processing was applied to the ISMN sites and satellite grids, and neither their spatial difference nor their representativeness was considered in this article. The ISMN multilayer in situ soil moisture measurements were used as a reference for comparison with SMOS and SMAP soil moisture products [45,46]. Although potentially accompanied by the spatial mismatch, this type of absolute difference could turn into a relative difference similar to a "system bias" when all products were compared to the same reference object.

The second step was temporal matching. All five types of data have UTC timestamps but in different formats. Timing can be adjusted to the nearest time by rounding minutes and seconds to hours. No additional processing was required as the sampling rate of the in situ data is hourly. The timestamps of the SMOS L2, SMOS-IC, and SMAP L2 products include minutes and seconds, which were rounded to the nearest hour. The timestamp of the SMAP L4 product corresponds to the center of the 3 h averaging interval; therefore, it was mapped to this 3 h time set in a left-closed and right-open fashion. It can be assumed that the SMAP L4 product is complete on the hourly time axis as there was an estimate for each hour; the in situ data were nearly complete except for a small number of missing (invalid) records; and the SMOS L2, SMOS-IC, and SMAP L2 products were discrete due to their temporal resolution.

There were two temporal matching schemes. The first was matching the in situ data with the satellite products one at a time. This type of comparison was expected to independently reflect the numerical characteristics of the satellite soil moisture. The second was matching all data simultaneously, which can be considered as eliminating the influence of the temporal mismatch and therefore allowing a comparison between satellite products [47,48]. The sample size of each matching group is shown in Table 1. It should be noted that timestamp is only one of the auxiliary information and cannot be utilized to discuss the temporal representativeness and rationality of the products.

**Table 1.** Sample size of temporal matching groups.

| | | Temporal Matching Groups | | | Counts |
|---|---|---|---|---|---|
| ISMN | SMOS L2 | | | | 128,619 |
| ISMN | | SMOS-IC | | | 86,646 |
| ISMN | | | SMAP L2 | | 123,635 |
| ISMN | | | | SMAP L4 | 3,257,075 |
| ISMN | SMOS L2 | SMOS-IC | SMAP L2 | SMAP L4 | 7848 |

2.2.3. Analysis of the Vertical Distribution Characteristics of Surface Soil Moisture

The overall distribution of soil moisture in each layer can be represented by its mean value. According to the maximum record of the ISMN data and the nominal retrieval accuracy of SMOS and SMAP products, the detailed distribution can be expressed by the segmented statistics of sample size in the total range of 0~0.52 $m^3/m^3$. The distribution analysis was based on all samples without distinguishing the site to which they belong.

The similarity of soil moisture between the layers can be quantified by the (Pearson) correlation coefficient ($R$) [49,50]. Three correlation sets were formed, namely, 5/5.08 and 10/10.16 cm (5/5.08 and 10/10.16); 10/10.16 and 20/20.32 cm (10/10.16 and 20/20.32); and 20/20.32 and 5/5.08 cm (20/20.32 and 5/5.08). The correlation coefficient was calculated separately for each site and was also presented in groups according to the static variables (land cover, sand fraction, and clay fraction), which were designed to reflect, to some extent, the potential influence of external environmental factors on the vertical distribution of soil moisture. It should be noted that the correlation coefficient only indicates the similarity

between the two sets of samples from a numerical point of view and cannot explain the coupling mechanism of soil moisture between the layers.

The difference in soil moisture between the layers can be directly expressed by their actual numerical differences, and the detailed distribution can also be reflected by the segmented statistics of sample size. Three sets were formed, namely, 5/5.08 minus 10/10.16 cm (5/5.08 − 10/10.16); 10/10.16 minus 20/20.32 cm (10/10.16 − 20/20.32); and 5/5.08 minus 20/20.32 cm (5/5.08 − 20/20.32). The soil moisture difference was calculated for all samples without distinguishing the site to which they belonged. The positive and negative differences were counted separately, as well as the average and the total number of samples on both sides.

2.2.4. Comparison between the Satellite Products and the In Situ Data

The comparison between the SMOS/SMAP products and the ISMN data was carried out on the basis of temporal matching (Table 1), using the actual numerical difference as an indicator to present the difference between them. Similarly, segmented statistics of sample size were used to visualize the detailed distribution of the differences. The positive and negative differences were counted separately, as were the mean and total sample sizes on both sides. The mean difference (*MD*) and mean absolute difference (*MAD*) were used as quantification indices according to the following equations:

$$MD = \frac{\sum(satellite - in\_situ)}{sample\ size}. \tag{1}$$

$$MAD = \frac{\sum|satellite - in\_situ|}{sample\ size}. \tag{2}$$

**3. Results**

*3.1. Stratification Characteristics of Surface Soil Moisture*

3.1.1. Single-Layer Distribution

As shown in Figure 2, there seemed to be a turning point at 0.24~0.28 m$^3$/m$^3$. For the 5/10/20 cm group, when it was below this range, the distribution of 5 and 10 cm showed some similarity. With an increase in depth, the peaks of the three layers gradually moved to higher ranges (0.04~0.08, 0.08~0.12, 0.16~0.20 m$^3$/m$^3$), especially in the ranges of 0~0.04 m$^3$/m$^3$ and 0.16~0.20 m$^3$/m$^3$, and the low-value characteristics of 5 cm and the high-value characteristics of 20 cm were very significant. However, above this range, a strong similarity was found between 10 and 20 cm, and the distribution difference among the three layers was reduced. For the 5.08/10.16/20.32 cm group, the sample size ranking of the three layers showed opposite trends below and above the inflection point; the distribution of 5.08 and 10.16 cm was also found to be similar, and their peaks were both located around 0.20~0.24 m$^3$/m$^3$. The distribution of 20.32 cm was very different from the other two layers, as its peak appeared at 0.32~0.40 m$^3$/m$^3$ where the soil moisture was very high. Although the difference in depth was small, the soil moisture of the two groups behaved quite differently; their means indicated that the soil moisture of the 5.08/10.16/20.32 cm group was always slightly higher than that of the 5/10/20 cm group (0.178/0.196/0.200 vs. 0.200/0.223/0.244 m$^3$/m$^3$). However, both showed a pattern of increasing soil moisture with the increase in depth, which appeared to be a stable distribution state of soil moisture.

The mean values of soil moisture in each layer were compared in groups according to the static variables of land cover, sand fraction, and clay fraction, and the results are shown in Figure 3. The general trends of the two sets of curves appear to be similar at first sight. For the 5/10/20 cm group (Figure 3A), 5 cm soil moisture showed a significantly low-value characteristic; the mean values of 10 and 20 cm soil moisture were very close, but the latter was slightly higher. Regardless of the static variables, the order of the three soil moisture layers from low to high remained unchanged with the increase in depth. For the 5.08/10.16/20.32 cm group (Figure 3B), 5.08 cm soil moisture was still the lowest, the

difference between 10.16 cm and 20.32 cm became larger, and in some cases, 10.16 cm soil moisture was higher.

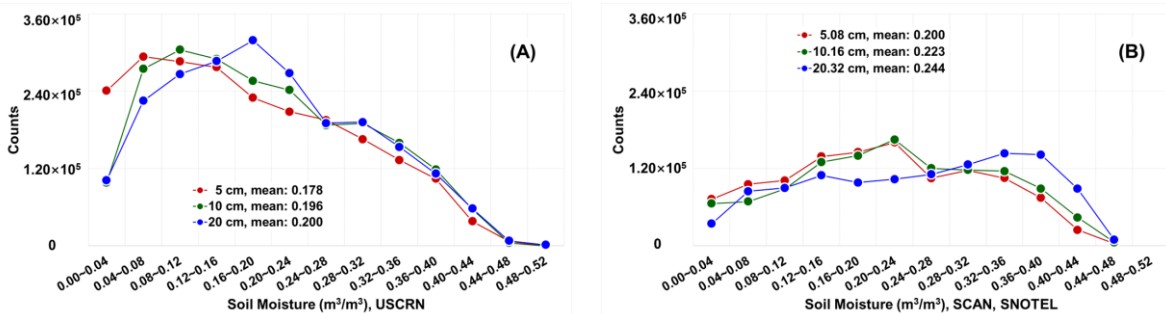

**Figure 2.** Soil moisture distribution in each layer: (**A**) the distribution at 5, 10, and 20 cm; (**B**) the distribution at 5.08, 10.16, and 20.32 cm (2, 4, and 8 in).

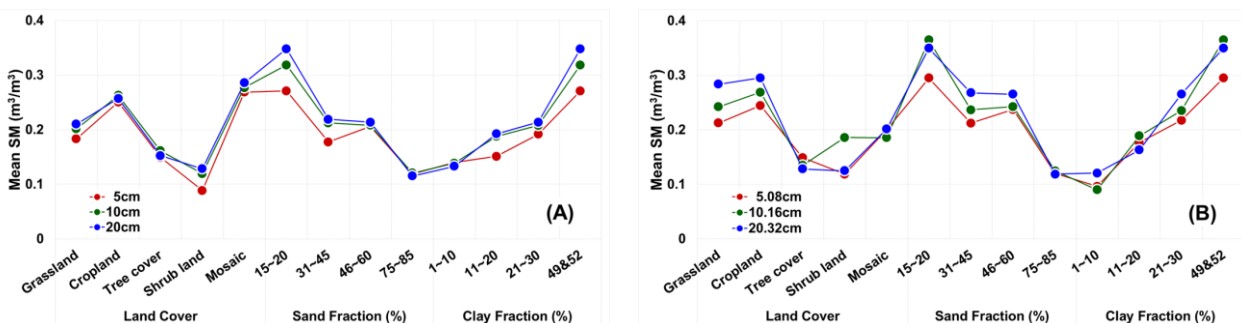

**Figure 3.** Mean soil moisture under static variables: (**A**) for the 5/10/20 cm group; (**B**) for the 5.08/10.16/20.32 cm group (2, 4, and 8 in).

The difference between the two groups was most obvious in terms of land cover. The 5.08/10.16/20.32 cm group showed stronger stratification characteristics than the 5/10/20 cm group in grassland, cropland, and shrubland conditions. Although the means of the three layers were close within each group under the conditions of tree cover and mosaic (mainly multiple vegetation types), there were large differences between the two groups.

As the sand and clay fractions increased, soil moisture tended to decrease and increase, respectively. The three layers differed significantly within and between the two groups, especially the pair of 10.16 and 20.32 cm. It appeared that the difference in the vertical distribution of soil moisture between the three layers became smaller with the increase in sand and larger with the increase in clay. For the 5/10/20 cm group, the influence of soil properties was slightly higher than that of land cover, but both types of static variables played a significant role for the 5.08/10.16/20.32 cm group.

### 3.1.2. Interlayer Correlation

The correlation coefficients of soil moisture between layers were calculated for each site, and the results are shown in Figure 4. The two groups showed a common pattern, i.e., the correlation coefficients decreased with increasing depth difference, although those of the 5/10/20 cm group were higher than those of the 5.08/10.16/20.32 cm group (0.899/0.884/0.813 vs. 0.809/0.767/0.690), and their distributions appeared to be very different.

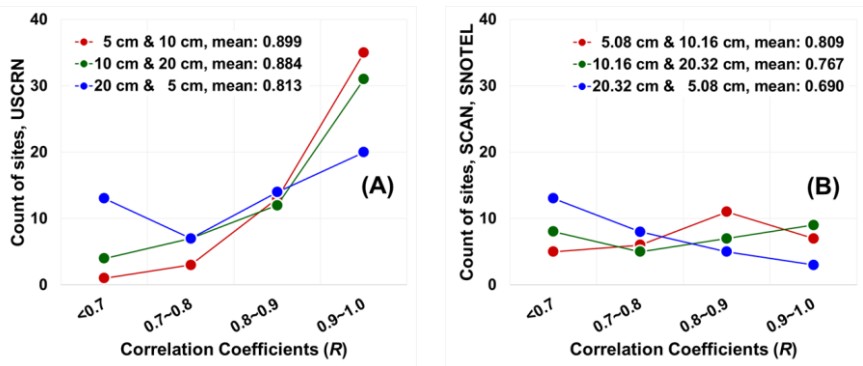

**Figure 4.** Correlation coefficients of soil moisture between layers: (**A**) interlayer correlation coefficients for the 5/10/20 cm group; (**B**) interlayer correlation coefficients for the 5.08/10.16/20.32 cm group (2, 4, and 8 in).

For the 5/10/20 cm group (Figure 4A), the distribution of the correlation coefficients of the three sets all showed an upward trend. Taking 0.8~0.9 as the turning point, in areas where the correlation coefficient was below 0.8, the order of the number of sites from small to large was "5&10", "10&20", and "20&5"; in areas where the correlation coefficient was above 0.9, the order was reversed. For the 5.08/10.16/20.32 cm group (Figure 4B), the three sets were distributed differently and no uniform trend was found. The downward trend of "20.32&5.08" looked very significant, while both "5.08&10.16" and "10.16&20.32" had an upward trend, although their peak and trough were located at 0.8~0.9 and 0.7~0.8, respectively. However, it can still be seen that the number of sites of "5.08&10.16" was highest in the intervals where the correlation coefficient was high, that of "20.32&5.08" was highest in the intervals where the correlation coefficient was low, and that of "10.16&20.32" always remained in the middle of the other two sets. This also reflected, to some extent, the tendency for the interlayer correlation coefficient to decrease as the depth difference increased.

The correlation coefficients were also grouped according to static variables, as shown in Figure 5. Firstly, in most cases, the order from lowest to highest was still "20/20.32&5/5.08", "10/10.16&20/20.32", and "5/5.08&10/10.16", with the difference between the three sets also increasing as the depth difference increased. Secondly, the correlation coefficients of the 5/10/20 cm group were always higher than those of the 5.08/10.16/20.32 cm group, except for the conditions of mosaic and the 75~85 sand fraction. Thirdly, for the 5/10/20 cm group (Figure 5A), the distribution of "20&5" appeared quite different from the other two sets, especially in shrubland, the 15~20 sand fraction, and the "49&52" clay fraction; there was not much difference between "5&10" and "10&20", and they were almost the same in some conditions. For the 5.08/10.16/20.32 cm group (Figure 5B), the three sets were quite different from the 5/10/20 cm group. They seemed to have a good synchronous trend, but the differences were very pronounced in the land cover condition.

It can be observed that the correlation coefficients of soil moisture between the layers decreased with the increase in depth difference, where the depth difference was 5/5.08 cm (10/10.16–5/5.08), 10/10.16 cm (20/20.32–10/10.16), and 15/15.24 cm (20/20.32–5/5.08). However, this could only indicate that the vertical similarity of soil moisture is related to the depth difference, but it was not possible to confirm where the depth difference lay. The correlation coefficients of "5/5.08&10/10.16" might not be the highest if the in situ measurements of 15/15.24 cm were provided, as there would be two more sets of depth differences also equal to 5/5.08 cm (15/15.24–10/10.08 and 20/20.32–15/15.24).

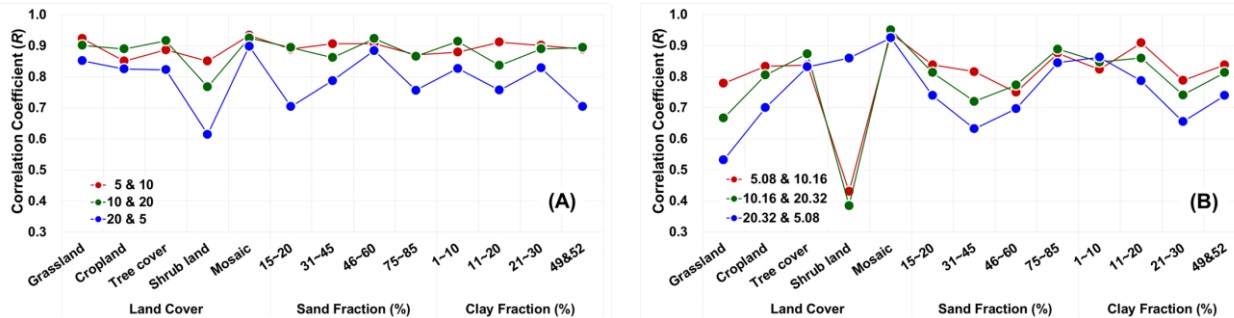

**Figure 5.** The correlation coefficients grouped according to the static variables: (**A**) the average correlation coefficients for the 5/10/20 cm group; (**B**) the average correlation coefficients for the 5.08/10.16/20.32 cm group (2, 4, and 8 in).

### 3.1.3. Interlayer Difference

As shown in Figure 6, the two groups reflected two types of vertical distribution in terms of mean and sample size for both the negative and positive values. For the 5/10/20 cm group, the order of the negative difference from small to large was "10–20", "5–10", and "5–20", indicating that the soil moisture of 10 cm was close to that of 20 cm, and the soil moisture difference between 5 cm and the other two lower layers (10 and 20 cm) increased with the increase in depth difference ($-0.32$, $-0.42$ m$^3$/m$^3$). The positive difference showed a consistent increasing trend (0.020, 0.024, and 0.028 m$^3$/m$^3$), but the sample size was much smaller than that of the negative difference; perhaps it can be assumed that this reverse increase with distance between the layers was random rather than conventional and was probably caused by precipitation. For the 5.08/10.16/20.32 cm group, the negative difference between the layers became more significant ($-0.044$, $-0.048$, and $-0.067$ m$^3$/m$^3$), with the sample size on both sides, showing a consistent trend of increase and decrease. The cases where the upper soil moisture was higher than the lower can also be explained by the influence of precipitation. The basic characteristics of soil moisture increasing with depth were more pronounced and showed a uniform variation in the vertical direction.

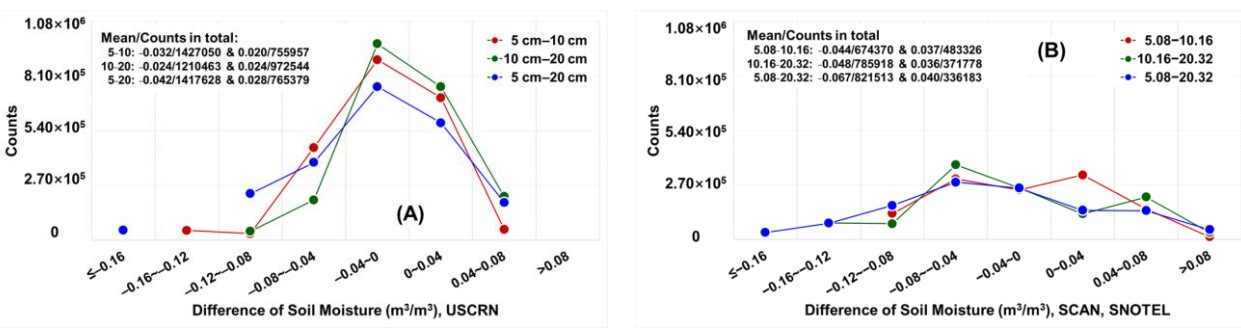

**Figure 6.** Soil moisture difference between layers: (**A**) the interlayer difference for the 5/10/20 cm group; (**B**) the interlayer difference for the 5.08/10.16/20.32 cm group (2, 4, and 8 in).

In terms of detailed distribution, for the 5/10/20 cm group, the peaks of the three sets of differences were all within $-0.04\sim0$ m$^3$/m$^3$; the upper limit of the positive differences was the same and did not exceed 0.04~0.08 m$^3$/m$^3$, while the lower limit of the negative differences was inconsistent, with the order from small to large being "10–20 ($-0.12\sim-0.08$ m$^3$/m$^3$)", "5–10 ($-0.16\sim-0.12$ m$^3$/m$^3$)", and "5–20 ($\leq-0.16$ m$^3$/m$^3$)". The 5.08/10.16/20.32 cm group seemed to be spread over a wider range than the other group, with the peaks moving backward to around $-0.08\sim-0.04$ m$^3$/m$^3$; the maximum positive difference was all above 0.08 m$^3$/m$^3$, and the descending order of the minimum nega-

tive difference became "5–10 ($-0.12\sim-0.08$ m$^3$/m$^3$)", "10–20 ($-0.16\sim-0.12$ m$^3$/m$^3$)", and "5–20 ($\leq-0.16$ m$^3$/m$^3$)".

The mean positive and negative differences under static variables and the difference in sample size between the two sides were shown in Figure 7. The difference in soil moisture between the layers at the two sets of depth was very different (Figure 7A,B) and was more pronounced in cases of low vegetation (grassland, cropland, and shrubland), lower sand fraction (15~20 and 31~45), and higher clay fraction (21~30, 49, and 52). Consistent with the difference in soil moisture, the difference in sample size appeared to be greater among the static variables (Figure 7C,D). The difference in soil moisture between the layers can be more clearly distinguished not only within each group but also between the two groups, further demonstrating the influence of land cover and soil properties on the water-holding capacity. It is worth mentioning that in terms of soil property, although the magnitude of the two sets of soil moisture difference was very different, the overall trend was similar. The sand and clay fractions given by ISMN refer to the soil property of 0~30 cm, and if we focus only on the surface layer of 0~5 cm, the difference in the composition may not be large; in other words, the soil property may not be the main factor affecting the vertical distribution characteristics of shallow soil moisture. Based on years of "big data", it may be possible to model the behavior of soil moisture under normal and disturbed conditions to provide more straightforward optimization solutions for soil moisture retrieval algorithms (brightness temperature simulation, parameter modeling, ancillary information assimilation, etc.).

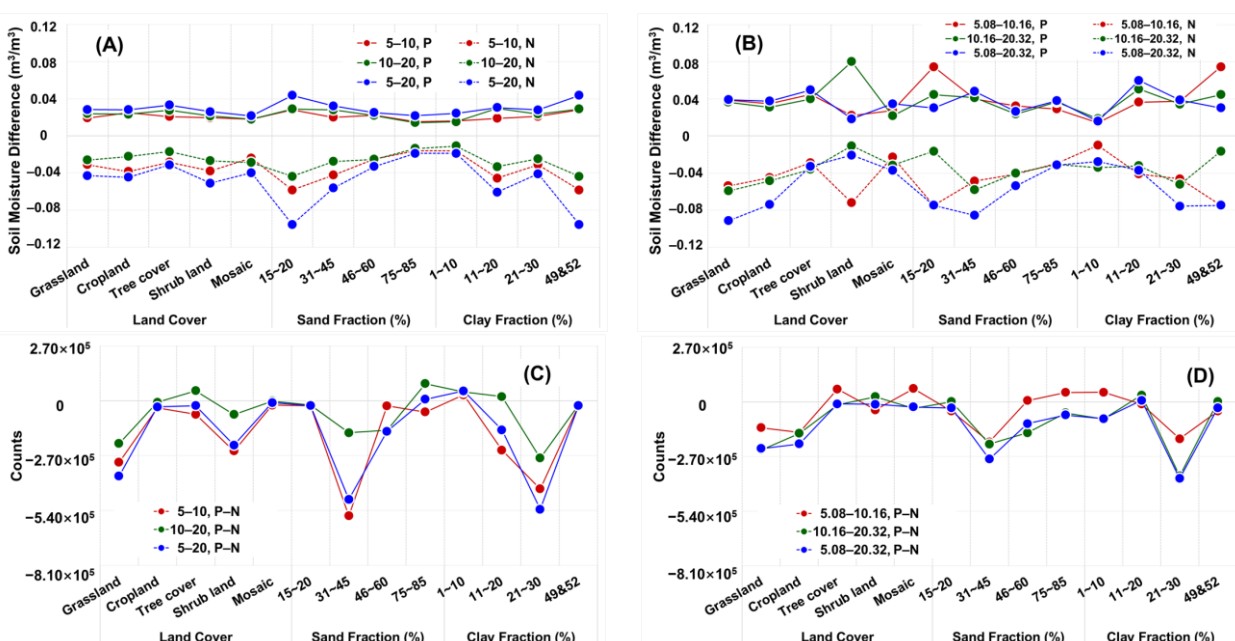

**Figure 7.** The interlayer difference of soil moisture and sample size under static variables; P and N refer to positive and negative, and P-N refers to positive minus negative: (**A**) the interlayer difference for the 5/10/20 cm group; (**B**) the interlayer difference for the 5.08/10.16/20.32 cm group (2, 4, and 8 in); (**C**) the difference of the sample size between the positive and negative sides for the 5/10/20 cm group; (**D**) the difference of the sample size between the positive and negative sides for the 5.08/10.16/20.32 cm group (2, 4, 8 and in).

### 3.2. Comparisons between the Satellite Products and the In Situ Data

The comparison was carried out in two ways based on temporal matching (Table 1). The first was comparing each type of satellite product separately with the three-layer in situ data (SMOS L2/SMOS-IC/SMAP L2/SMAP L4—in situ), and the second was comparing all four types of satellite products simultaneously with each single-layer in situ data (satellite products—5/5.08/10/10.16/20/20.32 cm).

### 3.2.1. Separate Comparison

The correlation coefficients are shown in Table 2. For within groups, it decreased with the increase in depth. For between groups, the correlation coefficients with the 5/10/20 cm group were slightly higher than those with the 5.08/10.16/20.32 cm group, and for the satellite products, the order from small to large was SMOS L2, SMOS-IC, SMAP L2, and SMAP L4. It can be seen that the satellite soil moisture products correlate better with the 5/5.08 cm in situ data than with the other two layers.

**Table 2.** Correlation coefficient of satellite soil moisture products and multilayer in situ measurements, separate comparison.

| *R* | 5 cm | 10 cm | 20 cm | 5.08 cm | 10.16 cm | 20.32 cm |
|---|---|---|---|---|---|---|
| SMOS L2 | 0.461 | 0.510 | 0.397 | 0.462 | 0.334 | 0.404 |
| SMOS IC | 0.675 | 0.607 | 0.610 | 0.559 | 0.538 | 0.493 |
| SMAP L2 | 0.648 | 0.629 | 0.586 | 0.580 | 0.524 | 0.500 |
| SMAP L4 | 0.701 | 0.654 | 0.655 | 0.613 | 0.602 | 0.572 |

As shown in Figure 8, each satellite product had its own unique performance. For SMOS L2 (Figure 8A), the peaks of its difference with the in situ data were around $-0.04\text{\textasciitilde}0$ (5 cm), $-0.1\text{\textasciitilde}-0.04$ (5.08, 10, 20 cm), and $-0.2\text{\textasciitilde}-0.1$ m$^3$/m$^3$ (10.16, 20.32 cm), reflecting, to some extent, the dry bias referred to in the literature. For SMOS-IC (Figure 8B) the peaks of the difference shifted to $-0.2\text{\textasciitilde}-0.1$ m$^3$/m$^3$ for all but 5.08 cm ($-0.1\text{\textasciitilde}-0.04$ m$^3$/m$^3$), implying an improved dry bias. For SMAP L2 (Figure 8C), the peaks of its difference with 5, 5.08, and 10 cm in situ data were around $0\text{\textasciitilde}0.04$ and $0.04\text{\textasciitilde}0.1$ m$^3$/m$^3$, where the dry bias started to change to a wet bias. For SMAP L4 (Figure 8D), the difference around $0.1\text{\textasciitilde}0.3$ m$^3$/m$^3$ was suppressed, and the wet bias was weakened, while the difference around $-0.1\text{\textasciitilde}0$ m$^3$/m$^3$ was enhanced, and the dry bias was strengthened.

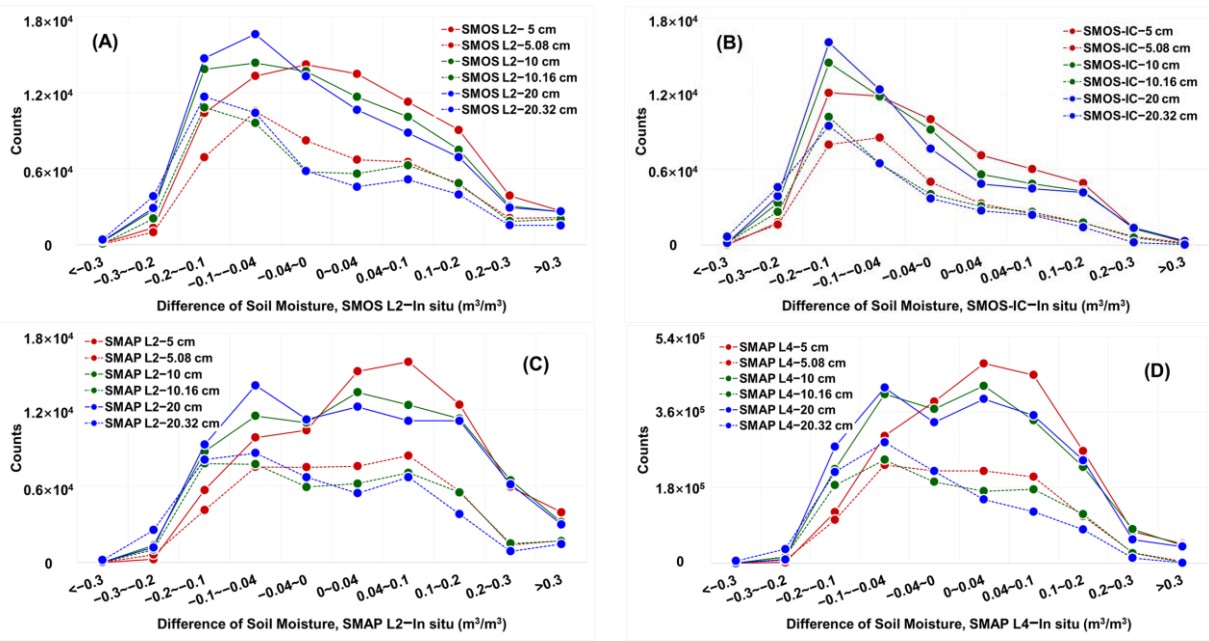

**Figure 8.** Soil moisture difference between satellite and in situ, separate comparison: (**A**) the difference between SMOS L2 and in situ; (**B**) the difference between SMOS-IC and in situ; (**C**) the difference between SMAP L2 and in situ; (**D**) the difference between SMAP L4 and in situ.

There seemed to be a turning point in the distribution of the difference between the four satellite soil moisture products and the three layers of in situ data. For the 5/10/20 cm

group, the turning point was around $-0.04 \sim 0$ m$^3$/m$^3$ except for SMOS-IC, and for the 5.08/10.16/20.32 cm group, the turning point was around $-0.1 \sim -0.04$ m$^3$/m$^3$ except for SMAP L4. The order of sample size from large to small was 20/20.32, 10/10.16, and 5/5.08 cm in areas where the difference was below the inflection point, while above the inflection point, the order was reversed. In general, the difference between the satellite products and the 5/5.08 cm in situ data was not similar to the other two layers, with SMOS L2 and SMOS-IC soil moisture lower than the in situ data and SMAP L2 and L4 soil moisture higher than the in situ data.

The numerical difference between the satellite products and in situ data was further explored in groups. The first group was based on land cover, sand fraction, and clay fraction. For each condition, the mean positive and negative difference was calculated separately, as well as the difference in sample size on both sides, and the results are shown in Figure 9.

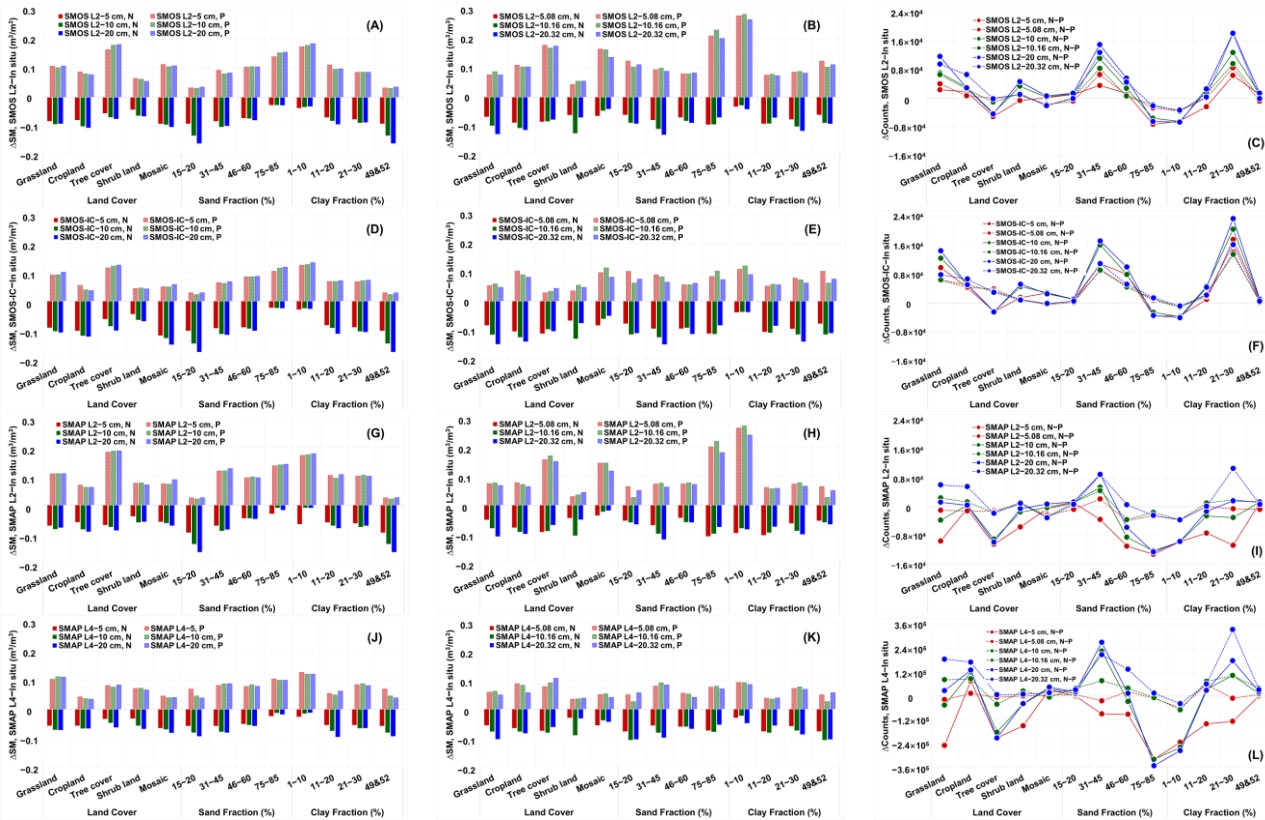

**Figure 9.** (**A**–**L**) Soil moisture difference between satellite and in situ, grouped by land cover, sand fraction, and clay fraction, separate comparison; P and N refer to positive and negative, and N − P refers to positive minus negative.

When comparing the satellite products with the 5/10/20 cm in situ data, the difference was significantly different in the cropland, tree cover, and mosaic conditions. The largest negative and positive differences were observed for SMOS-IC in the mosaic condition (Figure 9D) and SMAP L2 in the tree cover condition (Figure 9G). The negative difference decreased, and the positive difference increased with the increase in the sand fraction, while this trend was completely reversed with the increase in the clay fraction. The largest negative difference was contributed by SMOS-IC in conditions with the "15~20" sand fraction and "49&52" clay fraction, and the largest positive difference was contributed by SMOS L2 in conditions with the 75~85 sand fraction and 1~10 clay fraction (Figure 9A).

When compared to the 5.08/10.16/20.32 cm in situ data, none of the differences between the four satellite products and the in situ data were similar, especially in the tree cover and mosaic conditions. The negative difference in SMOS-IC (Figure 9E) and the

positive difference in SMOS L2 (Figure 9B) appeared to be higher than those of the other products. The trend of increasing and decreasing negative and positive differences could still be found with variations in sand and clay fractions, but the pattern was not as clear and consistent. In conditions where the sand fraction was very high, and the clay fraction was very low, the positive difference in SMOS L2 and SMAP L2 (Figure 9H) increased to about 0.2~0.3 $m^3/m^3$, which can be considered anomalies. In conclusion, regardless of the group with which the comparison was carried out, the negative difference between the satellite products and 5/5.08 cm in situ data was the smallest, and it was the largest with 20/20.32 cm; however, a similar pattern of a positive difference could only be found for SMOS L2, SMOS-IC, and SMAP L2 with their comparison to the 5/10/20 cm group.

The difference in sample size shown in Figure 9 is also revealing. Compared with the 5/10/20 cm in situ data, the sample size distributions of the four satellite products looked very different in the grassland condition but appeared similar in the mosaic condition. SMOS-IC (Figure 9F) and SMAP L4 (Figure 9L) were similar in the cropland condition, with a significantly higher negative than positive sample size, whereas in the tree cover and shrubland conditions, the sample size bias showed similarities within the SMOS (SMOS L2 and SMOS-IC) and SMAP (SMAP L2 and SMAP L4) groups, as well as differences between the groups. In terms of soil properties, the negative bias gradually became positive as the sand fraction increased, whereas the opposite trend was observed as the clay fraction increased, with exceptions where the sand fraction was very low (15~20) and the clay fraction was very high (49&52). Compared with the 5.08/10.16/20.32 cm in situ data, SMOS-IC in cropland and tree cover conditions and SMAP L4 in grassland conditions appeared to be significantly different from the other satellite products. The shift in dominance was still clearly discernible as the sand and clay fraction increased, and its magnitude slowed down but became more uniform for the SMAP group (SMAP L2, L4). There was a general pattern in which the difference between negative and positive sample sizes increased with depth.

The second group was based on the in situ data, and the results are shown in Figure 10. The mean negative difference increased with the increase in soil moisture, and the order from largest to smallest was SMOS-IC, SMOS L2, SMAP L2, and SMAP L4. The satellite products had the smallest negative difference with the 5/5.08 cm in situ data and the largest with the 20/20.32 cm. The descending order of the positive difference was SMOS L2, SMAP L2, SMOS-IC, and SMAP L4. A trend of decreasing positive difference with the increase in soil moisture can be found for SMAP L2 and SMAP L4, especially when comparing SMAP L2 and 5.08/10.16/20.32 cm in situ data (Figure 10J). However, SMOS L2 and SMOS-IC did not show such a trend, and the peak of their positive difference occurred mainly around 0.3~0.4 $m^3/m^3$, where the soil moisture was at a higher level. In most cases, the positive difference between the satellite products and 5/5.08 cm in situ data was the smallest.

As the soil moisture increased, the difference in sample size showed a basic pattern in which the negative difference gradually exceeded the positive one, peaking at about 0.3~0.4 $m^3/m^3$. The sample sizes on both sides became comparable when the soil moisture was higher than 0.4 $m^3/m^3$, but their difference remained positive. However, the comparison with the 20 cm in situ data seemed to be quite different from the others, as the difference between negative and positive values reached a maximum at around 0.1~0.2 $m^3/m^3$, and then the gap between the two sides narrowed with the increase in soil moisture, but it did not cross the 0 line. The performance of SMOS-IC was also somewhat peculiar in that the difference in sample size remained above the 0 line (excluding 20 cm), which meant that the magnitude of the negative difference was always greater than that of the positive one. In contrast, SMOS-IC had the least variation in the difference in sample size, while SMAP L4 had the most; if the degree of variation in the difference was to be ranked from small to large, the order was 5/5.08, 10/10.16, and 20/20.32 cm.

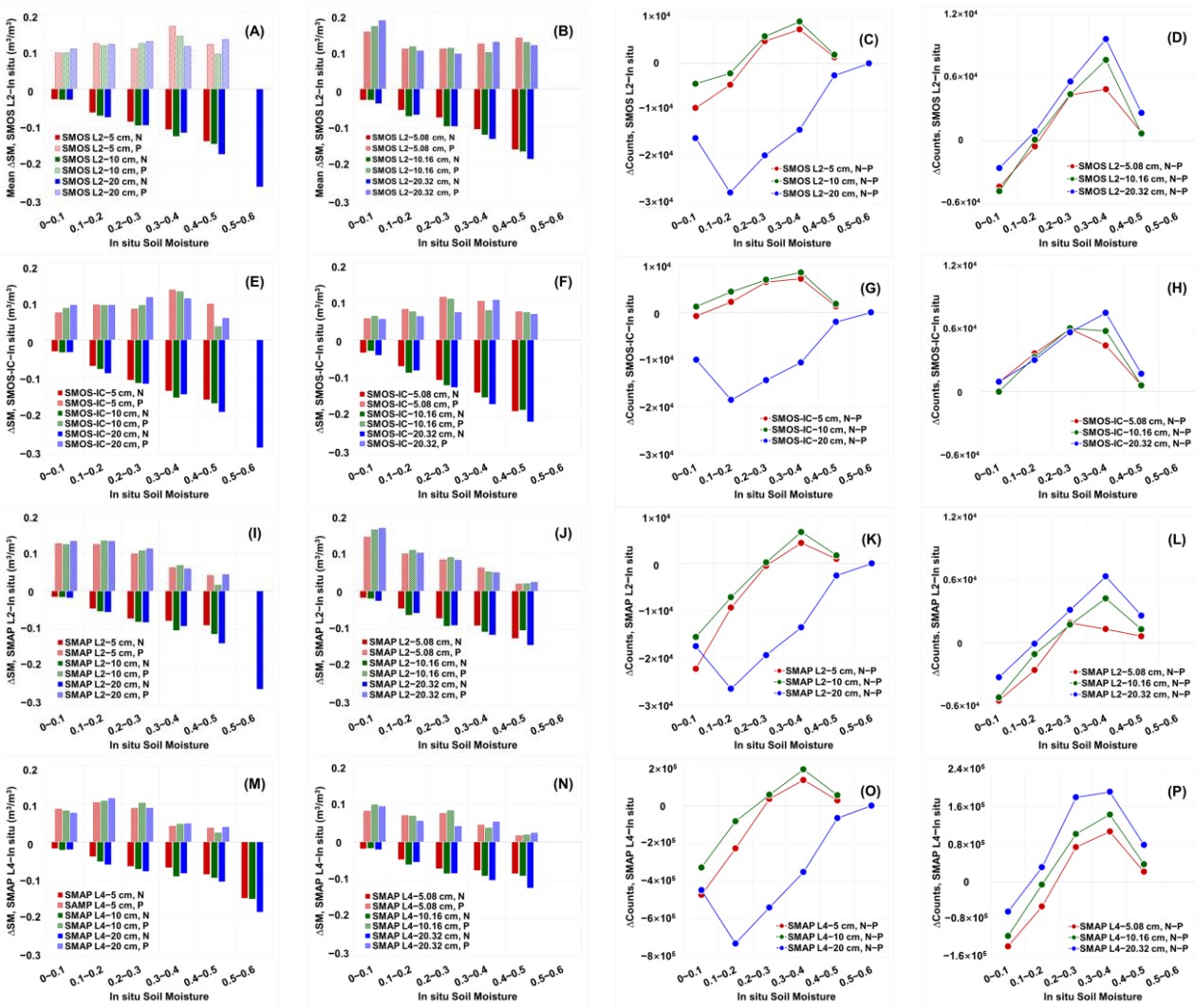

**Figure 10.** (**A–P**) Soil moisture difference between satellite and in situ, grouped according to in situ soil moisture, separate comparison; P and N refer to positive and negative, and N − P refers to positive minus negative.

### 3.2.2. Simultaneous Comparison

The SMOS L2, SMOS-IC, SMAP L2, and SMAP L4 soil moisture products were simultaneously compared with the in situ data at 5, 5.08, 10, 10.16, 20, and 20.32 cm, and their correlation coefficients and numerical differences are shown in Table 3 and Figures 11–13. It should be noted that the representativeness of the results may be limited, as the sample size was only 7848 under strict temporal matching (Table 1).

**Table 3.** Correlation coefficients of satellite soil moisture products and multilayer in situ measurements, simultaneous comparison.

| *R* | 5 cm | 10 cm | 20 cm | 5.08 cm | 10.16 cm | 20.32 cm |
|---|---|---|---|---|---|---|
| SMOS L2 | 0.535 | 0.557 | 0.463 | 0.479 | 0.381 | 0.453 |
| SMOS IC | 0.685 | 0.614 | 0.608 | 0.549 | 0.510 | 0.519 |
| SMAP L2 | 0.692 | 0.647 | 0.617 | 0.592 | 0.519 | 0.555 |
| SMAP L4 | 0.700 | 0.693 | 0.635 | 0.629 | 0.541 | 0.623 |

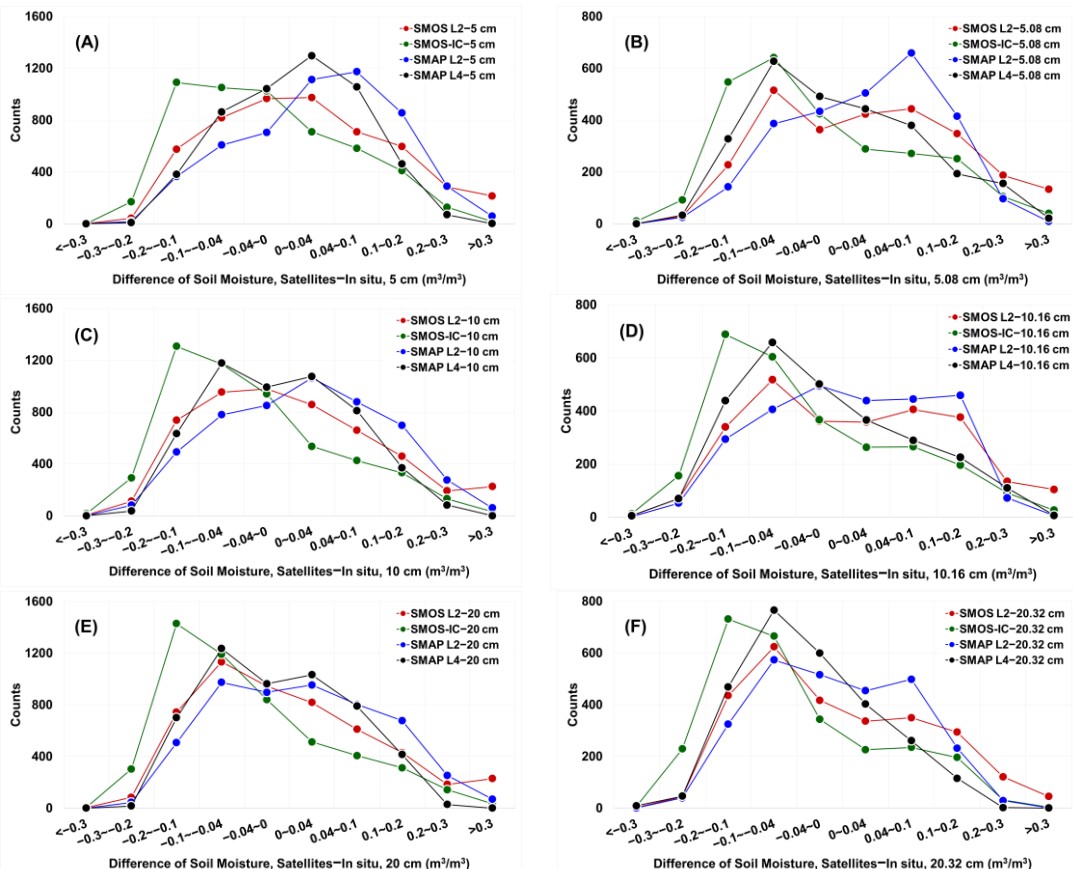

**Figure 11.** (A–F) Soil moisture differences between satellite and in situ data, simultaneous comparison.

The trends of the separate comparison (Table 2) are also presented in Table 3. Within the groups, the correlation coefficient decreased with the increase in depth. Between the groups, the SMOS-IC, SMAP L2, and SMAP L4 products had a higher correlation coefficient with the 5/10 cm in situ data than with the 5.08/10.16 cm. The ranking of the satellite products from small to large remained SMOS L2, SMOS-IC, SMAP L2, and SMAP L4, but they all had a higher correlation coefficient with the 5/5.08 cm in situ data.

The numerical difference between the four satellite products and the in situ data of each layer is shown in Figure 11, which shows the characteristics of each satellite product more clearly.

Compared with the 5 cm in situ data (Figure 11A), for SMOS L2, the difference concentrated within $-0.1 \sim 0.1$ m$^3$/m$^3$, and the negative was slightly higher than the positive. For SMOS-IC, the difference concentrated within $-0.2 \sim 0.04$ m$^3$/m$^3$, there was a peak around $-0.2 \sim -0.1$ m$^3$/m$^3$, and the negative was much higher than the positive. The difference for SMAP L2 seemed to be the opposite of SMOS-IC: It concentrated within $-0.04 \sim 0.2$ m$^3$/m$^3$, and the peak was around $0.04 \sim 0.1$ m$^3$/m$^3$, with the positive difference significantly higher than the negative. SMAP L4 seemed to have a normal distribution, as the difference was concentrated within $-0.1 \sim 0.1$ m$^3$/m$^3$, with a peak around $0 \sim 0.4$ m$^3$/m$^3$, and the positive was slightly higher than the negative, probably due to some calibration of the simulation when the soil moisture was high.

Compared with the 5.08 cm in situ data (Figure 11B), the dry bias of SMOS L2 would probably disappear since the size of the positive difference exceeded the negative, while the dry bias of SMOS-IC seemed to become stronger, with the difference narrowly concentrated within $-0.2 \sim 0$ m$^3$/m$^3$, and the size of the negative difference much higher than the positive. For SMAP L2, the difference remained positive without weakening. SMAP L4 was also found to have a remarkable dry bias, with the negative difference taking over and peaking at around $-0.1 \sim -0.04$ m$^3$/m$^3$.

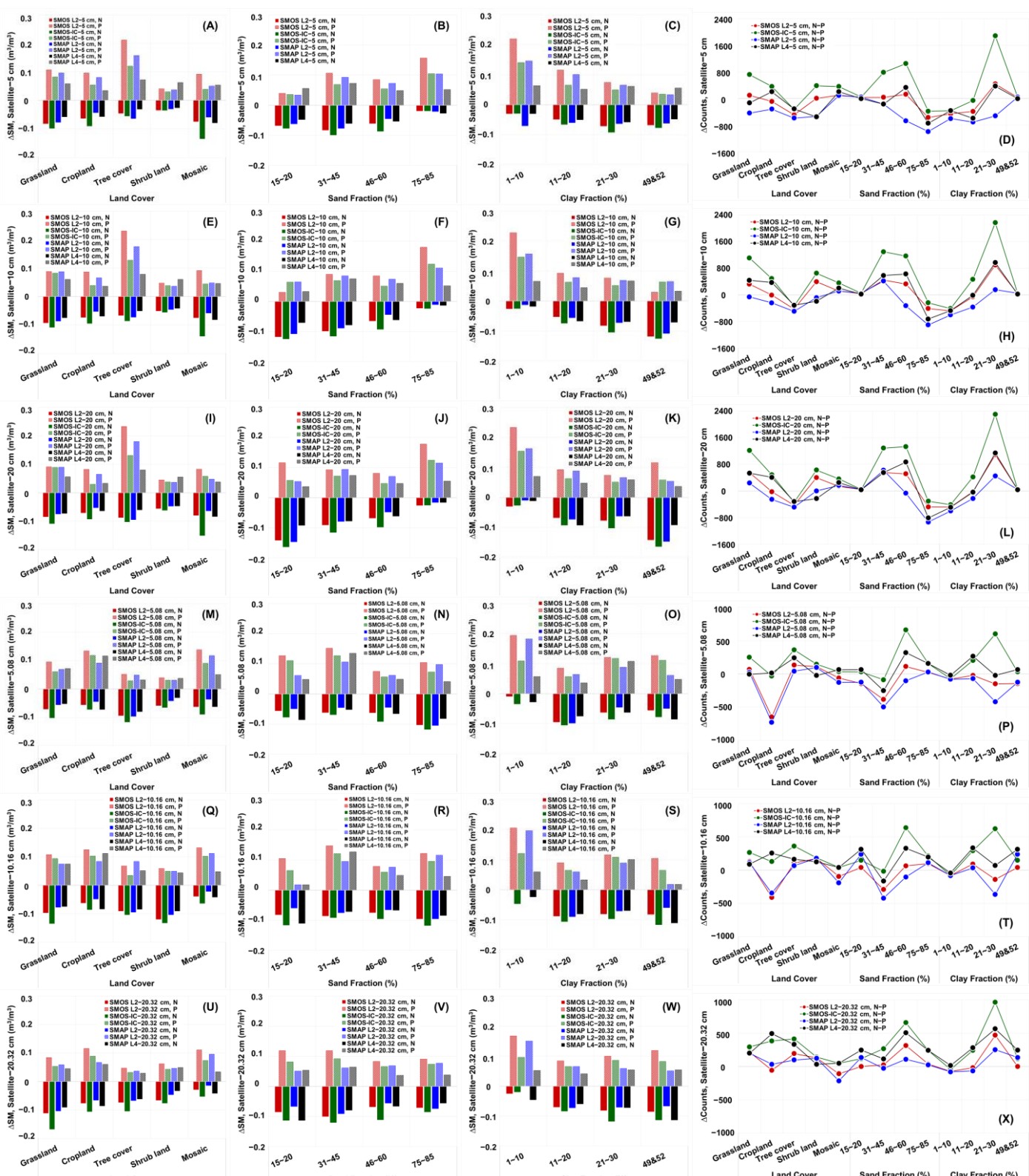

**Figure 12.** (**A**–**X**) Soil moisture differences between satellite and in situ data, grouped according to land cover and sand and clay fractions, simultaneous comparison; P and N refer to positive and negative, and N − P refers to positive minus negative.

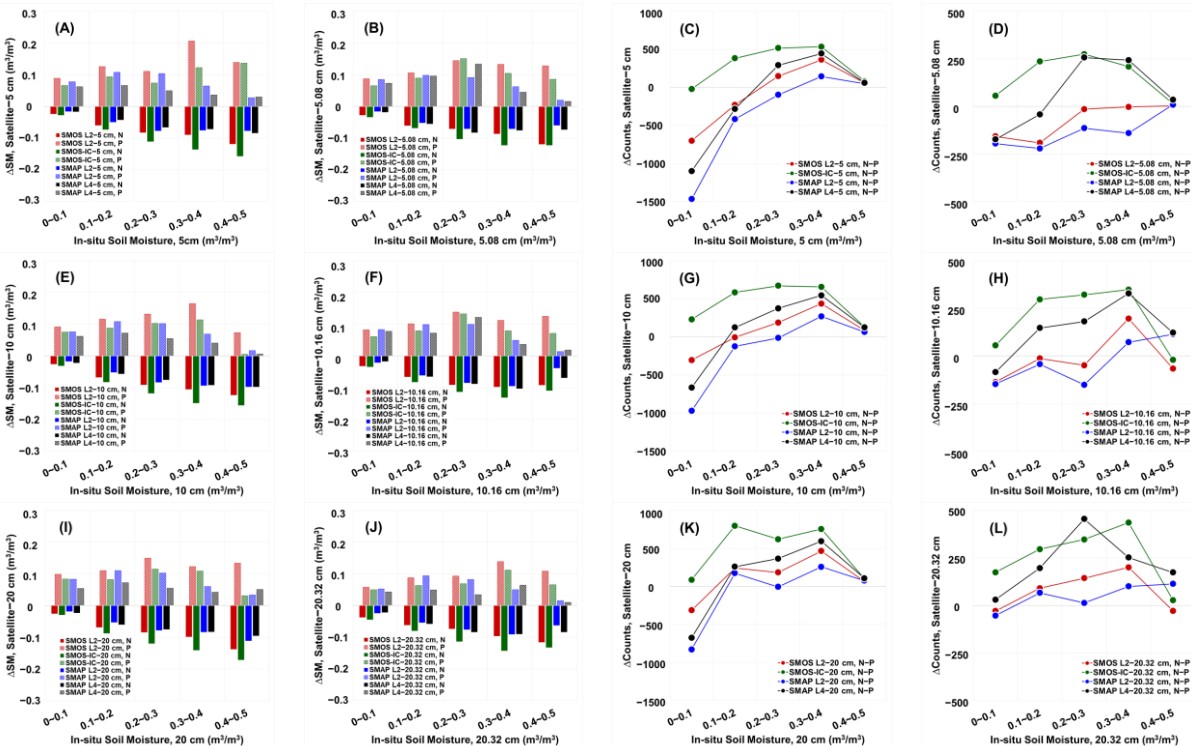

**Figure 13.** (**A–L**) Soil moisture differences between satellite and in situ data, grouped according to in situ soil moisture, simultaneous comparison; P and N refer to positive and negative, and N − P refers to positive minus negative.

Taking the comparison with the 5/5.08 cm in situ data as a reference, the differences between the four satellite products all moved progressively into the negative direction with the increase in depth, and the dry bias became stronger, and the distributions of their differences became more similar (Figure 11E,F). In addition, regardless of the depth to which the comparison was performed, the descending order of negative differences below the range of $-0.04\sim0$ m$^3$/m$^3$ was SMOS-IC, SMAP L4, SMOS L2, and SMAP L2, and when the differences were above this range, SMOS L2 had the largest scale of positive difference and SMOS-IC the smallest.

The differences between the satellite products and in situ data in a simultaneous comparison were also analyzed in terms of land cover, sand fraction, and clay fraction. The differences between the four satellite products varied in terms of land cover. In the comparison with the 5/10/20 cm in situ data (Figure 12A–L), the difference was largest in the tree cover and smallest in the shrubland, and there was little change in the positive difference with the increase in depth, but the negative difference gradually increased. The comparison with the 5.08/10.16/20.32 cm in situ data (Figure 12M–X) seemed to lack regularity, as there was a large negative difference in the tree cover, shrubland (Figure 12Q), and grassland (Figure 12U) but a large positive difference in the cropland and mosaic. The tendency for the negative difference to increase and the positive difference to decrease with the increase in depth could only be observed under grassland and cropland, with no common change for the others, and the comparison with the 10.16 cm in situ data seemed to show a large difference on both sides.

In the grouping of the sand and clay fractions, the trend in which the negative difference decreased and increased, respectively, as the two parameters increased remained highly significant compared with the 5/10/20 cm in situ data, and the opposite trend of the positive difference could also be distinguished. With the increase in depth, the negative difference continued to increase and reached a large magnitude with a low sand content and a high clay content (Figure 12J,K), while the positive difference was very high with a high sand content and a low clay content but did not show a clear pattern of variation

with depth. In comparison with the 5.08/10.16/20.32 in situ data, the trend of variation differed in each range of sand and clay fractions. There was a large negative difference in 31~45 (Figure 12V) and 75~85 (Figure 12N,R) sand fractions and a large positive difference in 31~45 sand fractions (Figure 12N,R) and 1~10 clay fractions, whereas the difference did not show a distinctive pattern of variation with sand and clay fractions but was found to increase in the negative difference and decrease in the positive difference with the increase in depth.

Of the four satellite products, SMOS L2 and SMOS-IC had the largest positive and negative differences, respectively, while SMAP L4 had the smallest positive and negative differences. The difference in sample size indicates that the deviation between the two sides can be arranged in descending order as SMOS-IC, SMAP L4, SMOS L2, and SMAP L2, with SMOS-IC mostly above the 0 line and SMAP L2 remaining below. Some cases are worth noting: Compared with the 5/10/20 cm in situ data (Figure 12D,H,L), SMAP L4, SMOS L2, and SMAP showed an increase and a reverse trend in the grassland and 31~45 sand fraction, and compared with the 5.08/10.16/20.32 cm in situ data (Figure 12P,T,X), there was a large decrease and a reverse trend in the cropland and 21~30 clay fraction. With the increase in depth, the distribution became closer to the 0 line and the fluctuation became weaker, which corresponds well to the trend in Figure 11 in which the magnitude of the negative difference increased and the predominance of the positive difference decreased.

The difference in depth between 5/10/20 cm and 5.08/10.16/20.32 cm was mainly due to the different unit settings of the observation depth, i.e., one was in centimeters and the other in inches. This 1.6% difference is difficult to detect in practice and may therefore be of little significance at a distance. The fundamental difference lies in the soil conditions and the type of land cover on which they rest, which will lead to not only an absolute difference between the networks but also a relative difference between stations within the network; in a sense, the difference between the satellite products and the two sets of in situ data may not be comparable. As mentioned before, land cover and soil properties are interdependent, and together, they drive the distribution characteristics of soil moisture in the vertical direction. The variety and variation in land cover in terms of temporal and spatial variables will probably be stronger and faster than those of the sand and clay fractions, and thus it has a greater influence on soil moisture. To some extent, this also indicates that the satellite retrieval of soil moisture should be more focused on land cover, especially the response and interaction with meteorological conditions of transient conditions.

The differences in the simultaneous comparison were also grouped according to the in situ data, and the results are shown in Figure 13. With the increase in soil moisture, the negative difference continued to increase, whereas the positive difference first increased and then decreased, peaking at around 0.3~0.4 $m^3/m^3$. SMOS-IC and SMOS L2 had the highest negative and positive differences, respectively, while SMAP L4 still remained the smallest on both sides. With the increase in depth, the negative difference showed an increasing trend, whereas most of the positive differences decreased. On the other hand, the distribution gradually approached or even crossed the 0 line with the increase in depth, indicating that the quantitative advantage of the negative difference constantly increased. SMOS-IC was above the 0 line and had a more negative difference, while SMAP L2 remained below this line and had a more positive difference, which is consistent with the results in Figure 11A,B and again confirms the numerical characteristics of the four satellite products.

### 3.2.3. The Depth Mismatch

To evaluate the depth mismatch, the mean difference (*MD*, Equation (1)) and mean absolute difference (*MAD*, Equation (2)) between the satellite soil moisture products and the multilayer in situ soil moisture data were calculated, and the results are presented in Tables 4 and 5.

**Table 4.** Differences between satellite data and in situ data: interlayer differences and separate comparison.

| (m³/m³) | MD | | | | MAD | | | |
|---|---|---|---|---|---|---|---|---|
| | **SMOS L2** | **SMOS-IC** | **SMAP L2** | **SMAP L4** | **SMOS L2** | **SMOS-IC** | **SMAP L2** | **SMAP L4** |
| Satellite−5 cm | 0.018 | −0.027 | 0.055 | 0.034 | 0.092 | 0.086 | 0.098 | 0.075 |
| Satellite−10 cm | −0.001 | −0.045 | 0.037 | 0.016 | 0.097 | 0.097 | 0.100 | 0.079 |
| Satellite−20 cm | −0.008 | −0.052 | 0.031 | 0.011 | 0.099 | 0.103 | 0.100 | 0.081 |
| (Satellite−10 cm) − (Satellite−5 cm) | −0.019 | −0.018 | −0.018 | −0.018 | 0.005 | 0.011 | 0.002 | 0.004 |
| (Satellite−20 cm) − (Satellite−10 cm) | −0.007 | −0.007 | −0.006 | −0.005 | 0.002 | 0.006 | 0 | 0.002 |
| (Satellite−20 cm) − (Satellite−5 cm) | −0.026 | −0.025 | −0.024 | −0.023 | 0.007 | 0.017 | 0.002 | 0.006 |
| Satellite−5.08 cm | 0.012 | −0.049 | 0.025 | 0.005 | 0.097 | 0.093 | 0.086 | 0.067 |
| Satellite−10.16 cm | −0.006 | −0.064 | 0.009 | −0.011 | 0.110 | 0.105 | 0.098 | 0.076 |
| Satellite−20.32 cm | −0.031 | −0.088 | −0.016 | −0.034 | 0.115 | 0.117 | 0.098 | 0.078 |
| (Satellite−10.16 cm) − (Satellite−5.08 cm) | −0.018 | −0.015 | −0.016 | −0.016 | 0.013 | 0.012 | 0.012 | 0.009 |
| (Satellite−20.32 cm) − (Satellite−10.16 cm) | −0.025 | −0.024 | −0.025 | −0.023 | 0.005 | 0.012 | 0 | 0.002 |
| (Satellite−20.32 cm) − (Satellite−5.08 cm) | −0.043 | −0.039 | −0.041 | −0.039 | 0.018 | 0.024 | 0.012 | 0.011 |

**Table 5.** Differences between satellite data and in situ data: interlayer difference and simultaneous comparison.

| (m³/m³) | MD | | | | MAD | | | |
|---|---|---|---|---|---|---|---|---|
| | **SMOS L2** | **SMOS-IC** | **SMAP L2** | **SMAP L4** | **SMOS L2** | **SMOS-IC** | **SMAP L2** | **SMAP L4** |
| Satellite−5 cm | 0.030 | −0.027 | 0.039 | 0.009 | 0.092 | 0.083 | 0.080 | 0.058 |
| Satellite−10 cm | 0.012 | −0.045 | 0.021 | −0.009 | 0.094 | 0.094 | 0.081 | 0.065 |
| Satellite−20 cm | 0.009 | −0.048 | 0.018 | −0.012 | 0.094 | 0.097 | 0.080 | 0.064 |
| (Satellite−10 cm) − (Satellite−5 cm) | −0.018 | −0.018 | −0.018 | −0.018 | 0.002 | 0.011 | 0 | 0.007 |
| (Satellite−20 cm) − (Satellite−10 cm) | −0.003 | −0.003 | −0.003 | −0.003 | 0 | 0.003 | −0.001 | −0.001 |
| (Satellite−20 cm) − (Satellite−5 cm) | −0.021 | −0.021 | −0.021 | −0.021 | 0.002 | 0.014 | 0 | 0.006 |
| Satellite−5.08 cm | 0.038 | −0.020 | 0.031 | 0.001 | 0.098 | 0.095 | 0.075 | 0.078 |
| Satellite−10.16 cm | 0.020 | −0.039 | 0.012 | −0.017 | 0.100 | 0.100 | 0.080 | 0.083 |
| Satellite−20.32 cm | −0.001 | −0.060 | −0.008 | −0.038 | 0.090 | 0.102 | 0.069 | 0.068 |
| (Satellite−10.16 cm) − (Satellite−5.08 cm) | −0.018 | −0.019 | −0.019 | −0.018 | 0.002 | 0.005 | 0.005 | 0.005 |
| (Satellite−20.32 cm) − (Satellite−10.16 cm) | −0.021 | −0.021 | −0.020 | −0.021 | −0.010 | 0.002 | −0.011 | −0.015 |
| (Satellite−20.32 cm) − (Satellite−5.08 cm) | −0.039 | −0.040 | −0.039 | −0.039 | −0.008 | 0.007 | −0.006 | −0.010 |

In the separate comparison, *MD* reflected the numerical characteristics of each satellite product well. It continued to grow in a negative direction with the increase in depth, regardless of whether it started out positive or negative. The dry bias of SMOS L2, the enhanced dry bias of SMOS-IC, the strong wet bias of SMAP L2, and the modified wet bias of SMAP L4 were clearly visible. The depth difference between 10 and 5 cm (−0.19~−0.18 m³/m³) was much larger than that between 20 and 10 cm (−0.07~−0.05 m³/m³), while the difference between 20.32 and 10.16 cm (−0.25~−0.23 m³/m³) was somewhat larger than that between 10.16 and 5.08 cm (−0.18~−0.15 m³/m³), also reflecting the stratification

characteristics of soil moisture. The *MAD* is actually the mean absolute cumulative difference, which increased slightly with depth. Focusing only on the first two layers, SMAP L4 always had the smallest *MAD*, while the largest *MAD* values were observed for SMAP L2 in the 5/10 group and SMOS L2 in the 5.08/10.16 group, respectively; the difference between 5/5.08 cm and 10/10.16 cm was slightly larger than that between 10/10.16 and 20/20.32 cm.

These results were further confirmed in the simultaneous comparison. *MD* also showed negative growth with depth, but the four satellite products behaved somewhat differently than in the separate comparison. SMOS L2 turned the dry bias into a wet bias, while SMAP L4 showed the opposite trend in the 5/10/20 cm group. SMOS-IC weakened the dry bias in the 5.08/10.16/20.32 cm group, and SMAP L2 weakened its wet bias in the 5/10/20 cm group. However, the interlayer difference remained stable, suggesting that, although the samples were screened in strict temporal matching, their inherent pattern did not change. *MAD* appeared to be slightly smaller in the simultaneous comparison, a ranking of the four satellite products could also be established, but there was still a lack of regularity.

## 4. Discussion

### *4.1. The Vertical Distribution Pattern of Surface Soil Moisture*

The stratification characteristics of soil moisture (5/10/20, 5.08/10.16/20.32 cm) were studied from three aspects: single-layer distribution, interlayer correlation, and interlayer difference. The fact that soil moisture in the upper layers was less than that in the lower layers seemed to be a stable distribution pattern, as the negative difference (upper–lower) dominated, and to some extent, this can be regarded as a natural response to gravity. The small increase in the mean positive difference ($0.020/0.024/0.028$ vs. $0.037/0.036/0.040$ m$^3$/m$^3$) should be noted, as it probably indicated that the soil moisture was close to or at saturation, in other words, that the maximum water capacity of this layer had been reached. The reverse growth reflected by the positive difference could be caused by external random conditions such as precipitation and can be considered an unconventional distribution pattern. Land cover and soil properties appeared to be the main determinants of the vertical distribution of soil moisture, particularly for shallow layers, where the effect of land cover may be greater. These two static variables were coupled and together determine the water-holding capacity of the soil. In conclusion, the absolute values of the positive and negative differences in soil moisture between the layers were very close to or even greater than $0.04$ m$^3$/m$^3$, indicating that there was significant stratification in the vertical direction and that the effect of depth mismatch on the validation and comparison of satellite soil moisture products should be carefully considered.

### *4.2. The Difference between the Satellite Products and the In Situ Data*

Land cover and soil properties of the sand and clay fractions were considered static variables and were used as the key parameters in the soil moisture retrieval algorithm. Quantification of the difference between the satellite soil moisture products and multilayer in situ measurements under these conditions is expected to provide references for data validation and algorithm optimization.

According to the separate comparison, the numerical difference showed that the satellite soil moisture retrievals had lower values than the in situ measurements. The dominance of the negative difference was likely to be the norm, and the background causing the positive difference could also be precipitation, as it occurred randomly and was mostly a persistent process, leading to an inverse distribution of soil moisture in the vertical direction. Such cases complicate the setting of dynamic conditions and ancillary information such as precipitation, temperature, and wetness, which in turn complicates the retrieval of soil moisture. Therefore, the retrieval optimization should more focus on soil moisture at higher levels, especially when the surface layer is high. It can be seen that the differences between all four satellite products and the 5/5.08 cm in situ data were smaller

than the differences between the four satellite products and 10/10.16 and 20/20.32 cm in situ data. A common pattern can be observed in which both the correlation coefficient and the numerical difference increased with the increase in depth.

In terms of simultaneous comparison, it is worth noting that, under each condition, namely, land cover, sand fraction, clay fraction, and soil moisture background, the difference between each satellite product varied with depth, but the order between them was roughly the same at all depths. In each of the products, unique strategies are used for setting these conditions in the soil moisture retrieval algorithm, which ultimately led to different results. The depth mismatch can be related to two aspects in the validation of the satellite products. The first was for the comparison between the satellite products and multilayer in situ data; their difference varied with depth, and the effect of the mismatch was observed. The second was for the comparison between the multisource satellite products; there was no significant change in the relative magnitude of their difference when they were all compared to the same in situ data at a given depth, and the mismatch effect may not be of concern.

In fact, the brightness temperature (TB, L1) was the common source of the soil moisture product at higher levels (L2 and L4). The reasons for the difference between the TB observations of SMOS and SMAP may be mainly due to their detection mechanism, hardware implementation, and reconstruction methods. However, the results of this study showed that the pattern of difference between the four satellite products and the multilayer in situ data did not change significantly with land cover, soil properties, and soil moisture background, which meant that the difference in penetration depth due to the observation conditions may not be large enough to cause the difference between the satellite soil moisture products.

## 5. Conclusions

Based on the ISMN multilayer in situ data (5, 10, 20, 5.08, 10.16, and 20.32 cm), the stratification characteristics of soil moisture were studied in this paper, and then SMOS (SMOS L2 and SMOS-IC) and SMAP (SMAP L2 and SMAP L4) soil moisture products were compared with the in situ data.

It was found that the soil moisture in the lower layers was usually higher than that in the upper layers, and there was a very significant hierarchical distribution in the vertical direction. The negative and positive differences of soil moisture between the layers were $-0.042/-0.67 \sim -0.024/-0.44$ and $0.020/0.036 \sim 0.028/0.040 \, \text{m}^3/\text{m}^3$, respectively, which were close to or even greater than the nominal retrieval accuracy of $0.04 \, \text{m}^3/\text{m}^3$ of SMOS and SMAP. The comparison showed that the correlation coefficient between the satellite products and the 5/5.08 cm in situ data was the highest, and their numerical difference was the smallest. The mismatch induced by using the 10/10.16 or 20/20.32 cm in situ data as a substitute was about $-0.019 \sim -0.018/-0.18 \sim -0.015 \, \text{m}^3/\text{m}^3$ and $-0.026 \sim -0.023/-0.043 \sim -0.039 \, \text{m}^3/\text{m}^3$ in the mean difference, respectively.

The mismatch of multisource data was mainly in the form of temporal, spatial, and depth mismatch. In previous studies, the influence of the temporal mismatch of SMOS and SMAP was found to be much smaller than the nominal retrieval accuracy of the satellites and can be safely ignored. The depth mismatch was analyzed in this study. It appeared to be larger than the temporal mismatch, according to the numerical differences.

Some shortcomings need to be mentioned. First, under the strict temporal matching, the sample size was too small to support a comparison of the sensitivity to the depth mismatch between satellite products. Second, the comparison between satellite products and multilayer in situ data was only formal, and their numerical differences could be due to multiple effects caused by external conditions such as precipitation, temperature, and wind, leaving much room for further research.

**Author Contributions:** Conceptualization, methodology, and formal analysis, N.Y.; software, validation, and investigation, F.X.; resources and data curation, H.Z.; writing—original draft preparation, N.Y.; writing—review and editing, F.X.; visualization, H.Z.; project administration and funding acquisition, N.Y. All authors have read and agreed to the published version of the manuscript.

**Funding:** This work was funded by the State Key Project of the National Natural Science Foundation of China—Key Projects of Joint Fund for Regional Innovation and Development (Grant Number U21A20108) and by the Double First-Class Project Cultivation Special Project—Key Technology for Intelligent Equipment and Intelligent Processing of Spatiotemporal Information (Grant Number 722403/067/004).

**Data Availability Statement:** No new data were created or analyzed in this study. Data sharing is not applicable to this article.

**Acknowledgments:** The authors would like to thank the International Soil Moisture Network (ISMN) for making available the field observations on soil moisture. We also wish to thank the European Astronomy Centre (ESAC) SMOS Data Processing Ground Segment (DPGS) for providing SMOS Level 2 data; INRAE BORDEAUX remote sensing products for providing SMOS-IC product; and the National Aeronautics and Space Administration (NASA) Distributed Active Archive Center (DAAC) at National Snow and Ice Data Center (NSIDC) for providing SMAP Level 2 and Level 4 data. We thank the Editorial Review Board Members and reviewers of *Remote Sensing*, for the time they took to review the manuscript and for their valuable feedback.

**Conflicts of Interest:** The authors declare no conflict of interest.

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
