# Peer review of "The Characterization of the Vertical Distribution of Surface Soil Moisture Using ISMN Multilayer In Situ Data and Their Comparison with SMOS and SMAP Soil Moisture Products"

_remotesensing, doi:10.3390/rs15163930_

Round 1

Reviewer 1 Report

Soil moisture is one of the key parameters in the Earth system, and validation of soil moisture product is essential before its application. The manuscript entitled “Vertical Distribution Characteristics of Surface Soil Moisture and the Depth Difference in the Comparison of SMOS and SMAP Soil Moisture Products” is a well-intentioned attempt to investigated the stratification characteristics of soil moisture using in-situ data from the ISMN networks (83 sites from 3 networks at 6 layers), and made comparisons with four L-band microwave remote sensing soil moisture products (SMOS L2, SMOS-IC, SMAP L2 and SMAP L4). The article is on an important topic and fits the scope of this journal.

The vertical distribution characteristics of surface soil moisture were analyzed in terms of single layer distribution, and correlation and difference between layers. The in-situ observations were adopted as the reference to evaluate the performance of the four soil moisture products in terms of land cover, soil texture (sand and clay fractions) and soil moisture conditions. The authors conclude that surface soil moisture has very pronounced stratification differences, that the SMAP L4 product shows better similarity to in-situ data, and that the depth mismatch between field measurements and satellite observations (retrievals) is a non-negligible effect in the validation. The topic is interesting, and the presented results of this study are helpful for the soil moisture community. However, the writing of this manuscript should be improved, the scientific novelty needs to be stren strengthened, and the conclusions require further clarification, a major revision is recommended, and the paper merits consideration for publication in RS after the detailed issues have been carefully addressed.

Abstract

The abstract needs to be revised. The problem under investigation should be better emphasised. The main aim of the paper should be clearly stated. The most important quantitative results should be clarified in the last part of the abstract.

1. Introduction

The introductory section is a bit short, more research on the literature should be done, and necessary references are needed in the SMOS and SMAP description part. Authors should better point out the scientific question that they addressed and what is new in this paper.

2. Materials and Methods

Authors should provide references to datasets that they mentioned. Methodology section lacks references and scientific explanation, the logic of this part needs to be improved.

3. Results and discussion

This part is a bit longer, the results and discussion seem to be mixed up. It is recommended to revise the organisation and structure to make each part more clear. For all the figures and tables: though the information in the figures and tables is interesting, their captions are a little simple. For example, what do the "N" and "P" mean in figures. 7, 9, 10, 12, 13? A detailed description of the captions of the figures and tables should be made.

4. Conclusions

The conclusions should be more focused, the answers to the key questions in the introduction need to be made more explicit.

Reference

Add the most recent references for the years 2022 and 2023.

N/A

Author Response

Dear Expert,

Thank you very much for your time and comments!

The authors have carefully revised the manuscript following your comments, please check the trace version of the manuscript.

Reviewer 2 Report

The article is interesting but some clarifications are needed

Author Response

Dear Expert,

Thank you so much for your time and comments! The authors have carefully revised and the notes are in the pdf file, please check it.

Round 2

Reviewer 1 Report

The authos have appropriately addressed my previous comments. I suggest to accept this manuscript.

Reviewer 2 Report

The authors have responded appropriately to my comments. I accept the paper for publication in its current version.